# Production of Modified Nucleosides in a Continuous Enzyme Membrane Reactor

**DOI:** 10.3390/ijms24076081

**Published:** 2023-03-23

**Authors:** Isabel Thiele, Heba Yehia, Niels Krausch, Mario Birkholz, Mariano Nicolas Cruz Bournazou, Azis Boing Sitanggang, Matthias Kraume, Peter Neubauer, Anke Kurreck

**Affiliations:** 1Department of Bioprocess Engineering, Institute of Biotechnology, Technische Universität Berlin, Ackerstr. 71-76, ACK24, 13355 Berlin, Germany; 2Department of Chemistry of Natural and Microbial Products, Pharmaceutical and Drug Industries Research Institute, National Research Centre, Dokki, Cairo 12622, Egypt; 3IHP—Leibniz-Institut für Innovative Mikroelektronik, Im Technologiepark 25, 15236 Frankfurt (Oder), Germany; 4DataHow AG, Hagenholzstrasse.111, 8050 Zurich, Switzerland; 5Department of Food Science and Technology, IPB University, Kampus IPB Darmaga, Bogor 16680, Indonesia; 6Department of Chemical and Process Engineering, Technische Universität Berlin, Straße des 17. Juni 135, 10623 Berlin, Germany; 7BioNukleo GmbH, Ackerstr. 76, 13355 Berlin, Germany

**Keywords:** continuous enzyme membrane reactor (EMR), halogenated nucleosides, nucleoside analogues, PID controller, purine nucleoside phosphorylase (PNP), pyrimidine nucleoside phosphorylase (PyNP), Raspberry Pi, thermophilic enzyme, transglycosylation

## Abstract

Nucleoside analogues are important compounds for the treatment of viral infections or cancers. While (chemo-)enzymatic synthesis is a valuable alternative to traditional chemical methods, the feasibility of such processes is lowered by the high production cost of the biocatalyst. As continuous enzyme membrane reactors (EMR) allow the use of biocatalysts until their full inactivation, they offer a valuable alternative to batch enzymatic reactions with freely dissolved enzymes. In EMRs, the enzymes are retained in the reactor by a suitable membrane. Immobilization on carrier materials, and the associated losses in enzyme activity, can thus be avoided. Therefore, we validated the applicability of EMRs for the synthesis of natural and dihalogenated nucleosides, using one-pot transglycosylation reactions. Over a period of 55 days, 2′-deoxyadenosine was produced continuously, with a product yield >90%. The dihalogenated nucleoside analogues 2,6-dichloropurine-2′-deoxyribonucleoside and 6-chloro-2-fluoro-2′-deoxyribonucleoside were also produced, with high conversion, but for shorter operation times, of 14 and 5.5 days, respectively. The EMR performed with specific productivities comparable to batch reactions. However, in the EMR, 220, 40, and 9 times more product per enzymatic unit was produced, for 2′-deoxyadenosine, 2,6-dichloropurine-2′-deoxyribonucleoside, and 6-chloro-2-fluoro-2′-deoxyribonucleoside, respectively. The application of the EMR using freely dissolved enzymes, facilitates a continuous process with integrated biocatalyst separation, which reduces the overall cost of the biocatalyst and enhances the downstream processing of nucleoside production.

## 1. Introduction

Nucleoside analogues are an important class of drugs. They have been widely used as anticancer, antibacterial, and antiviral medications [1]. The importance of this class of drugs is also evident in the current COVID-19 pandemic, as the nucleoside analogues molnupiravir and remdesivir were approved for the treatment of SARS-CoV-2 infections [2,3], and many more are currently under investigation in clinical trials [4]. Chemical synthesis is still the standard procedure for producing nucleoside analogues. The drawbacks, however, include numerous laborious reaction steps or the formation of various by-products, due to a lack of stereo- and regioselectivity, which necessitates the implementation of protection and deprotection steps [5,6]. Hence, often only low product yields are obtained, even after process optimization [7,8,9,10]. In contrast, chemo-enzymatic synthesis routes offer a suitable alternative, as biocatalysts are active in water-based reaction media and show high selectivity. Therefore, fewer process steps are required for both synthesis and purification, and the need for solvents is significantly reduced. Thus, major contributors to the E-factor (environmental factor, ratio of the mass of waste per mass of product) are minimized, compared to a wide range of chemical synthesis routes [11].

Nucleoside phosphorylases (NPs) are widely applied for the synthesis of modified nucleosides [1]. Based on their substrate spectrum, NPs are classified into purine nucleoside phosphorylases (PNP, EC 2.4.2.1) and pyrimidine nucleoside phosphorylases (PyNP, EC 2.4.2.2). Both catalyze the reversible phosphorolysis of nucleosides to nucleobases and α-D-pentofuranose-1-phosphates. Transglycosylation reactions are the most common approach to produce nucleoside analogues, which involve the interchange of a glycosyl moiety between a sugar donor (nucleoside) and a sugar acceptor (nucleobase) (Figure 1) [12,13]. In recent years, thermostable NPs have become increasingly interesting biocatalysts for nucleoside synthesis, as reactions at higher temperatures result in increased reaction rates and improve the solubility of substrates. Additionally, thermostable NPs have been reported to tolerate high solvent concentrations [14]. Hence, the use of higher substrate concentrations has led to increased volumetric yields, reduced E-factor values, and facilitated large scale application [1,11,14].

Providing the enzymes for enzymatic reactions on a large scale, is one of the most cost-intensive factors. Therefore, different methods have been developed that allow the reuse of active enzymes, or produce valuable products, in continuous processes. Enzyme immobilization has been most commonly used, and classically, enzymes have been bound to inert carrier materials [15,16,17,18,19,20,21]. Its successful application has already been demonstrated for nucleoside phosphorylases in several studies, for the synthesis of various natural and modified nucleosides [18,22,23]. Due to disadvantages of the classical immobilization techniques, including the use of expensive resins and large losses of enzyme activity due to immobilization, alternative methods have been developed in recent years. For example, covalent but reversible immobilization of a purine nucleoside phosphorylase on agarose microbeads, allowed the recycling of the resin after the enzyme became inactive [24]. In another approach, crosslinking-based self-immobilization of *Escherichia coli* uridine phosphorylase, even eliminated the need for an external carrier [25].

Despite the progress made, enzyme immobilization still leads to a significant loss of enzyme activity and to reduced productivity, caused by a limited mass transfer [26,27]. Enzyme membrane reactors (EMRs), using freely dissolved enzymes, offer a suitable alternative, as they combine the advantages of enzyme immobilization (e.g., enzymes are used until inactivation, easy separation of product and enzyme) and freely dissolved enzymes (e.g., maximum specific activity, no mass transfer limitations). In EMRs, enzymes are retained in the reactor by membranes with proper molecular weight cut-off (MWCO). EMRs are scalable and have already been applied to a large variety of products [28,29,30]. 

In this study, we tested an EMR system for the continuous synthesis of nucleoside analogues, using nucleoside phosphorylases of thermophilic origin as biocatalysts. We adjusted a previously described EMR system, that was successfully used for lactulose synthesis [27,31]. As it has been increasingly recognized that microelectronics can provide valuable technology modules for biotechnology [32], in this work the reactors were operated with an inexpensive and easy-to-handle proportional-integral-derivative controller (PID-controller), using a Raspberry Pi computer. The system was established and optimized for the enzymatic synthesis of deoxyadenosine. Afterwards, the reaction conditions were adjusted for the efficient synthesis of halogenated nucleoside analogues, as they serve as a valuable starting material for the preparation of a broad spectrum of nucleoside analogues [12,13,33].

## 2. Results and Discussion

### 2.1. Setup of the EMR

EMRs are widely used for the synthesis of a variety of compounds, in either single or cascade processes [34], but so far, no reports are available involving natural or modified nucleosides. Therefore, an automated system of two parallel EMRs was established for the synthesis of nucleosides, using a feedback controller. The membrane had an MWCO of <10 kDa, to retain the enzymes in the EMR. To achieve an operation at constant flux, and thereby a constant residence time, higher pressure is needed to compensate membrane fouling. The empirically determined PID parameters to achieve a constant flux were $K_{P}$ = 6, $K_{I}$ = 1.6, and $K_{D}$ = 3 (Figure 2B). After less than five minutes from the start of the EMR, accurate control of the permeate flux was achieved, with only minimal overshooting in the beginning (Figure 2B). In the further course of the EMR operation, stable control was achieved over long reaction periods.

### 2.2. Validation of the Functionality of the EMR Using 2′-Deoxyadenosine

To prove the functionality of the EMR controlled by the Raspberry Pi computer, the synthesis of 2′-deoxyadenosine (dAdo), in a transglycosylation reaction, was studied, using purified NPs. In this proof-of-concept study, products were not purified, and product yields were calculated from conversion percentages and volumes of product solutions. To estimate a suitable hydraulic residence time (HRT), batch reactions were performed at a 5 mL scale, over a period of 6 h. The one-pot bi-enzymatic reaction reached its thermodynamic equilibrium after 1 h, with dAdo formation and thymidine cleavage of 95% and 43%, respectively. In the thermodynamic equilibrium, the rates of thymidine (Thd) cleavage and dAdo formation were equal, and no apparent change in concentrations were observed anymore. Based on these results, it can be assumed that thermodynamic equilibrium can be reached at an HRT of 1 h, even in continuous mode. However, initial experiments showed that a minimum residence time of 4 h was required, because the setup was not designed to withstand the pressures generated at the required flow rates. Thus, an operational minimum of 4 h HRT was chosen for the initial dAdo synthesis, to avoid system leakage.

After setting up the EMR, by loading the substrates and enzymes, the reaction was started and run in batch mode for 1–2 h. In the initial batch phase, thymidine concentration decreased over time, while dAdo formation increased simultaneously (Appendix A). After equilibrium was reached, the formation of dAdo (95%) and cleavage of Thd (40%) was comparable to batch reactions at the 5 mL scale. Subsequently, continuous mode was started, with an HRT of 4 h, and dAdo formation and Thd cleavage remained constant compared to the batch mode. Thus, this residence time was sufficient to reach thermodynamic equilibrium for the formation of dAdo in the initial phase of EMR operation.

dAdo was successfully produced over a period of 55 days. Comparable longevity of nucleoside phosphorylases has previously been shown for immobilized *Halomonas elongata* PNP applied in a flow reactor [24]. With inosine and 6-O-methylguanine as substrates, the immobilized biocatalyst still showed significant activity after two months of usage for reactions in flow. Despite the longevity of the enzyme, a loss in activity over time was observed. After two months of application, the loss in activity was around 50%. A decrease in activity was also observed in the EMRs. Therefore, HRT had to be regularly adjusted, to ensure steady state conditions and dAdo formation at conversions >90%. Hence, after 22 days and 45 days, the HRT was increased to 6 h and 12 h, respectively (Figure 3A). Based on this procedure, a productivity of 2.5 g L^−1^ h^−1^ was reached for the EMR, which was in very good agreement with the 5 mL batch reactions. As the EMR was running for more than 50 days without the need for fresh enzymes, the continuous reaction resulted in 220 times more product per enzymatic unit than in the batch reaction (Table 1).

During the operation of the EMR, an increasing turbidity of the reactor content was observed (Figure 3B). The system pressure rose simultaneously, and critical values were reached (up to 4 bar). To ensure a continuous EMR performance, the process was stopped after 6, 14, and 23 days, to change the reactor membrane and to remove the formed precipitate by centrifugation (Figure 3A). SDS-PAGE analysis revealed that the precipitate mainly contained the applied biocatalysts (Appendix A). To ensure that solubility of the reactants did not lead to precipitate formation, a sample of the precipitate was also analyzed by high-performance liquid chromatography (HPLC). No peaks corresponding to the nucleosides or nucleobases in the reaction were observed. Denaturation of the biocatalysts, however, had only a minor effect on dAdo formation in the beginning. After three cycles of precipitate removal and membrane exchange, however, residence time was prolonged, to retain dAdo formation >90%.

In batch reactions, the formation of by-products was not observed, using thymidine and adenine as substrates. However, in the EMR, by-products started to form after 2 days of operation. A comparison with reference standards revealed the formation of 2′-deoxyinosine (dIno) and hypoxanthine (Hx), which are deamination products of dAdo and adenine (Ade), respectively. By-product concentrations increased with operation time and longer residence times, reaching a total of up to 5 mM final concentration (50% of the final product) at the end of the experiment (Appendix A). Deamination is a well-known challenge in enzymatic reactions and is mainly caused by enzyme impurities, originating from the heterologous expression host. The application of reaction temperatures above 60 °C [19,35] or adding deaminase inhibitors like 2′-deoxycoformycin, are suitable options to overcome this challenge [36].

### 2.3. Synthesis of Dihalogenated Nucleoside by Nucleoside Phosphorylases

As the EMR was running with good dAdo productivity, the synthesis of 2,6-dihalogenated nucleoside analogues was performed. To determine a suitable HRT for EMR operation, batch reactions to form 2,6-dichloropurine-2′-deoxyribonucleoside (2,6DCP-dR) and 6-chloro-2-fluoropurine-2′-deoxyribonucleoside (6C2FP-dR) were performed initially. As was also observed for the synthesis of dAdo, the reaction equilibrium was reached after only 1 h. Product conversions were 84% and 83% for 2,6DCP-dR and 6C2FP-dR, respectively, at equilibrium.

In the batch phase of the EMR runs, equilibrium was also reached within 1 h, with comparable conversion rates, of 86% and 84% for 2,6DCP-dR and 6C2FP-dR, respectively. Although an HRT of 4 h was used, due to the operational limitations of the EMR, 6C2FP-dR formation immediately decreased to 75% and remained constant for 6 h. Thereafter, the 6C2FP-dR formation ranged between 60 and 75%. The formation of 2,6DCP-dR did not show such fluctuations. Over a period of about 10 h, the 2,6DCP-dR formation was about 80%. Subsequently, it slowly started to decrease. After 50 h, product formation and Thd cleavage rates strongly decreased for both 2,6DCP-dR and 6C2FP-dR, probably caused by the inactivation of the biocatalysts. Hence, in comparison to the EMR run for dAdo synthesis, drastically reduced operation times were observed for the modified nucleosides. As HRT might be a critical factor for EMR operation, a prolonged HRT of 8 h was studied. Indeed, for both substrates, operation times increased with prolonged residence times. A significant decrease in product formation was only observed after 230 h and 62 h, for 2,6DCP-dR and 6C2FP-dR, respectively.

Productivities for small-scale batch reactions, and the EMR at an HRT of 8 h, were comparable, with values ranging between 1.3 and 1.6 g L^−1^ h^−1^ (Table 1). The values were in good accordance with results observed for the related compounds 2,6-dichloro-(ß-D-ribofuranosyl)purine (2,6DCP-R) and 6-chloro-2-fluoro-9-(ß-D-ribofuranosyl)purine (6C2FP-R) using immobilized NPs in batch reactions. Productivities of 1.5 and 2.0 g L^−1^ h^−1^ were observed for the synthesis of the two ribosides [15].

Product yields per employed enzymatic unit again increased remarkably in the EMRs, compared to the batch reactions. In the EMRs, with an HRT of 8 h, the product yields per enzymatic unit were 9 and 40 times higher, for 6C2FP-dR and 2,6DCP-dR, respectively (Table 1). As stated before, prolonged HRT had a positive effect on product yields for both dihalogenated products. However, the effect was much more pronounced for 2,6DCP-dR, where the product yields per employed enzymatic unit increased by a factor of 4.5, when increasing the HRT from 4 h to 8 h (Table 1).

The lifetime of the enzymes was found to be strongly dependent on the applied substrates. While the EMR to produce dAdo ran over a period of 55 days, operation times were much shorter for 6C2FP-dR and 2,6DCP-dR. Halogen substituents have previously been shown to interfere with enzyme activity or cell viability [37,38]. Halogenated nucleoside analogues such as 2-amino-6-chloro-7-deazapurine 2′-deoxyriboside and 6-amino-2-chloro-7-deazapurine 2′-deoxyriboside, were described as being PNP inhibitors, by forming a ternary dead-end PNP/base/P_i_ complex, while also being substrates in the reverse synthetic reaction [37]. Furthermore, nucleoside 2′-deoxyribosyltransferase, with a 2′-deoxy-2′-fluoroglycoside substrate, was found to form a destabilized oxocarbenium-like transition state, due to the electron-withdrawing fluorine atom at the sugar’s 2′ position, thus leading to accumulation of the covalently bound intermediate and thereby inactivating the enzyme [39]. Additionally, tissue or organ toxicity has been observed before for halogenated nucleoside analogues, such as 2-fluoroadenosine or fludarabine, in pharmaceutical applications [40,41,42,43,44]. Substitutions with reactive halogens at the purine ring, exert electron withdrawing centers, that can change the substrate orientation, affinity, pKa, stability, and impact adjacent groups [45].

## 3. Materials and Methods

### 3.1. Reactor Configurations

Two parallel EMRs were used to produce natural and modified nucleosides. The reactors were set up similarly to the description in Sitanggang et al. [27]. An EMR consists of a pressure-stable glass container and a holder, which was modified from a XFUF-047 dead-end test cell (Merck Millipore, Darmstadt, Germany). The maximum working volume was 92 mL. Flat sheet PES membranes, with a MWCO of 10 kDa (Microdyn Nadir, Wiesbaden, Germany), were placed at the bottom of the reactor. PTFE tubing (inner diameter (ID) = 0.8 mm) was employed to connect the substrate tank, enzyme reactor, and beakers, where the product solution was collected. The EMR was operated in continuous mode and samples were taken regularly, from a valve located between the enzyme reactor and product beaker. For permeate measurement, Kern precision balances (Kern & Sohn GmbH, Balingen-Frommern, Germany) were used. The proportional pressure regulator was purchased from Festo (MPPE-3-1/4-6-010-B, Esslingen am Neckar, Germany).

Instead of using the Laboratory Virtual Instrument Engineering Workbench (LabVIEW™ (National Instruments, US), as described by Sitanggang and colleagues, a PID controller (proportional-integral-derivative controller) was implemented in Python (version 2.7, Python Software Foundation, US), using a Raspberry Pi 3 B (Raspberry Pi Foundation, UK), with the PiXtend V2 -S- extension board, supplied by Qube Solutions GmbH (Germany).

The weight of the permeate passing through the membrane was measured on a precision balance, to calculate the deviation from the set hydraulic residence time: (1)$$HRT\;\left[h\right]=\frac{V_{R}}{\dot{V}}$$
where *V_R_* (L) is the reactor volume, and $\dot{V}$ (L h^−1^) the volumetric flow rate of the feeding solution. HRT describes the average duration of time that the reactants remain in the EMRs.

Process data (pressure, time, weight, target value) were simultaneously stored in a csv file and later used to calculate residence times and productivities. 

### 3.2. Control Design

To control the continuous process and ensure a steady conversion rate, a PID controller was developed, using a Raspberry Pi and the programming language Python. The controller measures the weight of the harvest tank with the collected product every six seconds and calculates the deviation (or error I) from the predefined target value, corresponding to the desired residence time. The output for the valve was calculated as follows:(2)$$\mathrm{Out}=K_{P}e\left(t\right)+K_{I}\int e\left(t\right)dt+K_{D}\frac{d}{dt}e\left(t\right)$$
where $e\left(t\right)$ is the deviation of the weight from the setpoint at the time point t, and $K_{P}$,$\;K_{I}$, and $K_{D}$ are the respective tuning parameters, reflecting a proportional, an integral, and a derivative term of the PID controller. The output was then converted, to adjust the pressure from 0–4 bar, to a value between 0 and 1023, the allowed range of the input for the valve, and sent to the valve via a serial interface. The process time, the target value, as well as the deviation and the set pressure, were repeatedly (every 6 s) saved to a csv file. 

The pressure was not allowed to exceed 4 bar, to ensure the integrity of the device. Therefore, the membrane was changed if needed, to continue operation within the acceptable pressure range. In case of insufficient conversion of the substrates, the setpoint of the residence time was manually increased and set accordingly.

The respective Python script is stored at the following git repository and freely available: https://git.tu-berlin.de/bvt-htbd/public/thiele_2023_ijms (uploaded on 9 January 2023).

### 3.3. Chemicals 

All chemicals and solvents were of analytical grade or higher and purchased, if not stated otherwise, from Sigma-Aldrich (Steinheim, Germany), Carl Roth (Karlsruhe, Germany), TCI Deutschland (Eschborn, Germany), Carbosynth (Compton, Berkshire, UK), or VWR (Darmstadt, Germany). High-performance liquid chromatography (HPLC) analyses were carried out with an Agilent 1200 series system, equipped with an Agilent diode array detector (DAD), using a Phenomenex (Torrance, CA, USA) reversed-phase C18 column (150 × 4.6 mm). Thermostable nucleoside phosphorylases PyNP Y02 (E-NP-1002) and PNP N02 (E-NP-2002) in purified form (immobilized metal ion-affinity chromatography), were provided by BioNukleo GmbH (Berlin, Germany) and used as recommended by the manufacturer.

### 3.4. Synthesis of Natural and Modified Nucleosides in Batch Reactions

The products 2′-deoxyadenosine (dAdo), 2,6-dichloropurine deoxyribonucleoside (2,6-DCP-dR), and 6-chloro-2-fluoropurine deoxyribonucleoside (6C2FP-dR) were synthesized in a one-pot transglycosylation reaction, using pyrimidine nucleoside phosphorylase PyNP 02 and purine nucleoside phosphorylase PNP 02 as biocatalysts. Thymidine was used as a sugar donor and adenine, 2,6-dichloropurine, and 6-chloro-2-fluoropurine as sugar acceptors. In a final volume of 5 mL, a reaction mixture of 25 mM Thd, 10 mM of sugar acceptor, 1.6 U mL^−1^ of PyNP 02, and 11.4 U mL^−1^ of PNP 02, in 2 mM potassium phosphate (KP) buffer (pH 7.0), was prepared. The reaction was incubated at 40 °C for 6 h. Regular samples were taken and analyzed by HPLC-DAD.

### 3.5. Operational Procedure to Produce Natural and Modified Nucleosides in Enzyme Membrane Reactors

To perform transglycosylation reactions, substrate concentrations of 25 mM Thd and 10 mM sugar acceptor (adenine, 2,6-dichloropurine, or 6-chloro-2-fluoropurine) were used, in both the initial batch solution and the feed. The substrates were dissolved in 2 mM KP buffer (pH 7.0). The syntheses were carried out using 150 U of PyNP 02 and 1050 U PNP 02, at a temperature of 40 °C. The agitation speed was 200 rpm.

After reactor assembly, stirring was started and the reactors were filled with substrate solution, up to 50 mL. Enzymes were added and the reactors were filled up to 92 mL with substrate solution. Reactions were run in batch mode until equilibrium was reached, after about 1.5 h. Thereafter, the continuous production of nucleosides and their analogues was performed with residence times of 4 h. If the system pressure reached almost 4 bar, the reactor membrane was exchanged, and the reactor content was centrifuged. The clear supernatant was transferred back into the reaction chamber. To maintain adequate conversion, HRT was increased up to 16 h, if necessary, during the process. 

The productivity of the batch and continuous processes was compared in terms of specific productivity, productivity/space-time-yield, and product yield per U of enzyme, and was calculated as follows:(3)$$\mathrm{Specific}\;\mathrm{productivity}\;\left[{\mathrm{mg}\;U}^{-1}h^{-1}\right]=\frac{\begin{array}{c} c_{product}\;\times V_{permeate} \end{array}}{U\;\times \Delta t\;\;}$$
with $c_{product}$ (mg L^−1^) being the product concentration, $V_{permeate}$ (L) being the permeate volume, U being the enzyme units, and Δ*t* (h) being the reaction period. For batch reactions, the reaction volume was 5 mL.

Productivity was calculated according to the equation: (4)$$\mathrm{Productivity}\;\left[{g\;L}^{-1}h^{-1}\right]=\frac{\sum_{t_{1}}^{t_{2}}P\left(t\right)}{\sum_{t_{1}}^{t_{2}}V\left(t\right)\times \Delta t}$$
where $\overset{t_{2}}{\underset{t_{1}}{\sum }}P\left(t\right)$ and $\overset{t_{2}}{\underset{t_{1}}{\sum }}V\left(t\right)$ are the amount of product and the permeate volume collected during a certain period *t_2_ − t_1_*, and Δ*t* the reaction period. 

The enzyme specific product yield was calculated as follows: (5)$$\mathrm{Product}\;\mathrm{yield}\;\mathrm{per}\;\mathrm{enzyme}\;\mathrm{unit}\;\left[{g\;U}^{-1}\right]=\frac{\sum_{t_{start}}^{t_{end}}P\left(t\right)}{U}$$
where $\overset{t_{end}}{\underset{t_{start}}{\sum }}P\left(t\right)$ is the total amount of product collected over the process time, and U the enzyme units in the reaction mixture.

### 3.6. Measurement of Enzyme Activity 

All reactions were performed at 2 mL scale, in Eppendorf tubes, which were incubated in a thermomixer (Eppendorf, Hamburg, Germany) at 40 °C and 300 rpm. The standard activity assay (phosphorolysis) was carried out in potassium phosphate buffer (50 mM; pH 7.0) containing 1 mM uridine or guanosine. After 2 min preheating at the corresponding temperature, 0.0003 mg mL^−1^ (final concentration) of purified enzyme was added to the mixture, and the reaction was stopped by the addition of ½ vol. methanol after defined time intervals, so that <10% of the substrate was converted to the product. Under these conditions, the reaction rate was linear as a function of time and enzyme concentration. After centrifugation (20,000× *g*, 4 °C, 15 min), the samples were analyzed by HPLC-DAD. One unit (U) of enzyme activity was defined as the amount of enzyme catalyzing the conversion of 1 µmol of substrate per minute, under the respective assay conditions.

### 3.7. Determination of Nucleosides and Nucleobases by HPLC 

HPLC-DAD analyses were performed to monitor the enzyme-catalyzed reactions. For the synthesis of dAdo, 2,6DCP-dR, and 6C2FP-dR, HPLC analysis was performed with the following gradient: from 97%, 20 mM ammonium acetate and 3% acetonitrile, to 72% 20 mM ammonium acetate and 28% acetonitrile, in 11 min. The conversion percentages were determined by measuring the nucleosides and nucleobases at 260 nm (Equation (5)). Substrates and products (including by-products) were identified by their retention times and specific spectra. Retention times were determined using pure compounds as standards. Under these conditions they were as follows: Thd (5.0 min), thymine (Thy) (4.2 min), dAdo (5.4 min), adenine (Ade) (4.4 min), 26DCP-dR (9.0 min), 2,6-dichloropurine (2,6DCP) (7.8 min), 6C2FP-dR (8.4 min), 6-chloro-2-fluoropurine (6C2FP) (7.1 min), hypoxanthine (Hx) (3.2 min), and 2′deoxy-inosine (dIno) (4.5 min). Reaction intermediates, such as phosphate and deoxyribose-1-phosphate, were not determined, as they are not freely available in the reaction, since transglycosylation reactions are a balanced interchange of the sugar moiety between the donor nucleoside and the acceptor. Hence, 2′-deoxyribose-1-phosphate released after donor nucleoside cleavage by PyNP, is immediately used by the PNP to form the nucleoside of interest. The inorganic phosphate released in the PNP-catalyzed reaction, is directly used for donor nucleoside cleavage.

To calculate conversion percentages, first, measured peak areas were converted into concentrations, based on standard curves obtained from HPLC measurements using defined concentrations of the pure compounds (substrates and products). Afterwards, conversion percentages were calculated, according to:(6)$$\mathrm{Conversion}\;\left[\%\right]=\frac{c_{product}\;\left[\mathrm{mM}\right]}{\begin{array}{c} c_{acceptor}\;\left[\mathrm{mM}\right]+c_{product\;}\left[\mathrm{mM}\right] \end{array}}\times 100$$
where $c_{product}$ is the concentration of the formed nucleoside, and $c_{acceptor}$ is the concentration of the residual sugar accepting nucleobase. Appendix A illustrates the calculation of the conversion using example chromatograms.

## 4. Conclusions

One of the major challenges for the development of industrial enzymatic processes, is the economic application of biocatalysts. Within the past decades, enzyme immobilization techniques have been widely applied, however, enzyme immobilization is often accompanied by a loss of activity. Here, we show that EMRs are a suitable alternative for the synthesis of natural and modified nucleosides, as they facilitated an efficient application of biocatalysts until their complete loss of activity. Due to the superior stability of thermostable enzymes, in the present study, the EMRs could run for up to seven weeks, depending on the applied substrate. Compared to the batch reactions, product yields per enzyme unit were significantly increased, by factors of 9, 40, and 220 for 6C2FP-dR, 2,6DCP-dR, and dAdo, respectively, after using prolonged residence times in the EMRs.

## Figures and Tables

**Figure 1 ijms-24-06081-f001:**
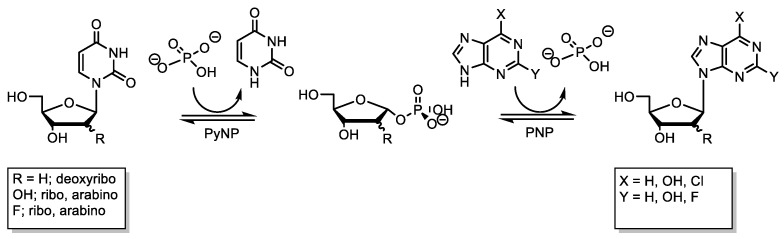
Enzymatic synthesis of modified nucleosides by nucleoside phosphorylase-catalyzed transglycosylation. For the synthesis of purine nucleosides, it is advantageous to use pyrimidine nucleosides as donors, as they are cheaply available and thermodynamically enhance purine nucleoside synthesis. In this bi-enzymatic reaction, the PyNP is needed to cleave the donor nucleoside and the PNP catalyzes the target nucleoside synthesis.

**Figure 2 ijms-24-06081-f002:**
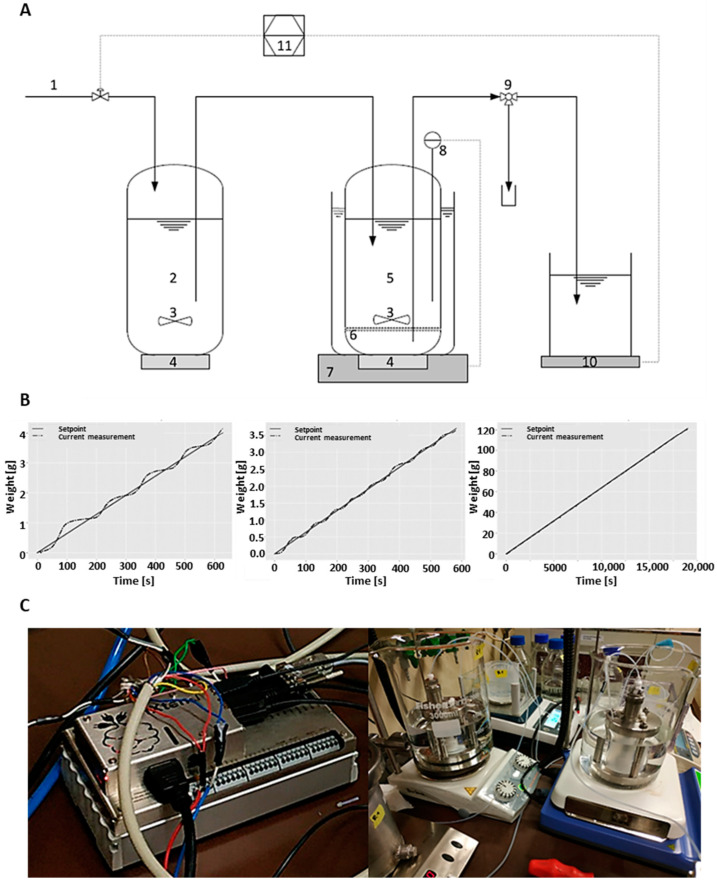
(**A**) Schematic representation of the experimental setup of a single enzyme membrane reactor. During the experiments, two enzyme membrane reactors were run in parallel (**C**). 1—Compressed air. 2—Substrate tank. 3—Magnetic stirrer. 4—Magnetic drives. 5—Enzyme reactor. 6—Ultrafiltration membrane. 7—Heating device. 8—Temperature sensor. 9—Sampling port. 10—Precision balance. 11—PID controller. (**B**) PID controller adjustment: screenshots of the tuning of the parameters to $K_{P}$ = 6, $K_{I}$ = 1.6, $K_{D}$ = 3 and correlation between the target value and measured value over time. Solid line—target value, dotted line—measured value. Tuning parameters were added subsequently from the left to the right panel. (**C**) Photo of the experimental setup of two parallel reactors during operation.

**Figure 3 ijms-24-06081-f003:**
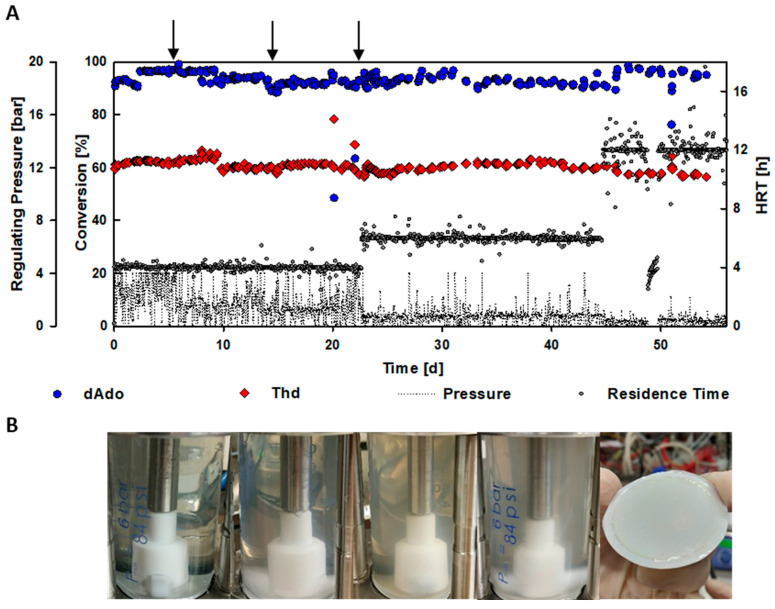
Continuous production of 2′-deoxyadenosine from thymidine (25 mM) and adenine (10 mM) using purified nucleoside phosphorylases (150 U of PyNP 02, 1050 U of PNP 02, by BioNukleo) at 40 °C, stirred at 200 rpm. (**A**) Monitoring of thymidine (Thd) cleavage and 2′-deoxyadenosine (dAdo) formation, and regulating pressure during operation, at hydraulic residence times (HRT) of 4, 6, and 12 h. Membrane exchange and centrifugation of the reactor content, to remove precipitate, is indicated by an arrow. (**B**) Turbidity of the reaction mixture inside the EMR and membrane clogging were observed over the course of the reaction.

**Figure 4 ijms-24-06081-f004:**
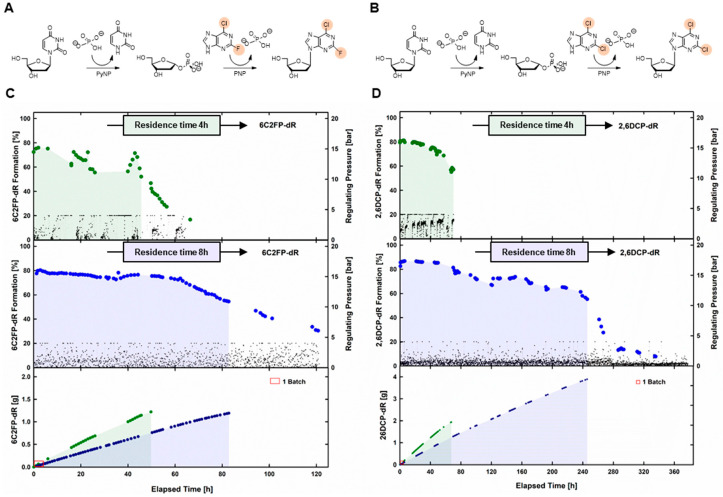
Continuous production of modified nucleosides from thymidine (25 mM) and 10 mM 6-chloro-2-fluoro-purine (**A**,**C**), or 2,6-dichloropurine (**B**,**D**), using purified pyrimidine and purine nucleoside phosphorylases (150 U of PyNP 02, 1050 U of PNP 02, by BioNukleo GmbH) at 40 °C, stirred at 200 rpm. Thymidine (Thd) cleavage and formation of the respective nucleoside at residence times of 4 and 8 h were monitored (**C**,**D**). The threshold value of conversion for the calculation of process parameters (Table 1) is indicated by the shaded area. Full data sets for the synthesis of the dihalogenated nucleosides are shown in Appendix A.

**Table 1 ijms-24-06081-t001:** Process parameters determined for the synthesis of dAdo and dihalogenated nucleosides. Parameters were calculated for steady state flow conditions in the EMRs. The threshold value of conversion was 55% substrate conversion for the calculation of process parameters for EMRs applied, to produce 6C2FP-dR and 2,6DCP-dR (see Figure 4).

Enzymes	Product	Productivity [g L^−1^ h^−1^]	Specific Productivity [mg U^−1^ h^−1^]	Product Per Enzymatic Unit [mg U^−1^]	Reaction System	Reference
PyNP 02/PNP 02	dAdo	2.49	0.22	1.32	Batch	This study
PyNP 02/PNP 02	dAdo	2.5	0.22	290.4 *	EMR	This study
PyNP 02/PNP 02	2,6DCP-dR	1.6	0.14	0.84	Batch	This study
PyNP 02/PNP 02	6C2FP-dR	1.4	0.12	0.72	Batch	This study
PyNP 02/PNP 02	2,6DCP-dR	1.5 1.6	0.13 0.14	7.8 (HRT 4 h) 33.6 (HRT 8 h/16 h)	EMR	This study
PyNP 02/PNP 02	6C2FP-dR	1.3 1.3	0.11 0.11	5.5 (HRT 4 h) 6.49 (HRT 8 h)	EMR	This study
TtPyNP ^a^/GtPNP ^b^	2,6DCP-R	1.5	-	-	Batch, immob. enzyme	[13]
TtPyNP/GtPNP	6C2FP-R	2.0	-	-	Batch, immob. enzyme	[13]

^a^ PNP of *Thermus thermophilus*, ^b^ PyNP of *Geobacillus thermoglucosidasius.* * During the EMR run, HRT was regularly adjusted to reach product formation > 90%.

## Data Availability

Detailed process data can be found in the Appendix A. The Python code is freely accessible via the following git repository: https://git.tu-berlin.de/bvt-htbd/public/thiele_2023_ijms (uploaded on 9 January 2023).

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
