# Peer review of "Production of Modified Nucleosides in a Continuous Enzyme Membrane Reactor"

_ijms, 2023, doi:10.3390/ijms24076081_

Round 1

Reviewer 1 Report

In this manuscript the authors describe a continuous procedure to synthesize modified nucleosides using nucleoside thermostable phosphorylases in an enzyme membrane reactor (EMR). In their reactions, thymidine is the sugar-donor substrate and acceptor nucleobases are adenine, and the halogenated-adenine derivatives 2,6-dichloropurine (2,6DCP) and 6-chloro-2-fluoropurine (6C2FP). The synthesis of the corresponding deoxynucleosides (dAdo, 2,6DCP-dR and 6C2FP-dR) was obtained in a one-pot, two-step enzymatic reaction. In the first one, pyrimidine nucleoside phosphorylase catalyzes the phosphorolysis of thymidine yielding thymine and 2’-deoxyribose-1-phosphate. In the second step, purine nucleoside phosphorylase catalyzes the glycosylation of 2’-deoxyribose-1-phosphate by one of the mentioned nucleobases, with liberation of phosphate. Globally, the bi-enzymatic process catalyzes a transglycosylation. So, the deoxynucleoside thymidine is converted to a deoxypurine compound, either dAdo, 2,6DCP-dR or 6C2FP-dR.

The authors are based in previous studies of bi-enzymatic batch synthesis of the same or similar compounds with phosphorylases, and in studies of the EMR use for the production of other types of compounds. Both kinds of previous expertise are here conjugated to compare the efficiency of phosphorylase transglycosylations by an EMR procedure versus similar reactions performed in a batch system. The aim of the authors is mainly demonstrative of EMR applicability for the synthesis of nucleoside derivatives, in which they succeed.

The manuscript will be relevant for those interested in the production of nucleoside analogs of pharmaceutical interest at industrial scale. However, there are several issues that need revision.

MAJOR ISSUES

1. The products formed are analyzed by HPLC monitored at 260 nm. This should detect substrates and (sub)products containing a nitrogen base, but not 2’-deoxyribose-1-phosphate and inorganic phosphate. However, no chromatogram is shown to illustrate the progress and endpoint of substrate consumption and (sub)product formation. This should be presented and used to explain how the percentages of thymidine consumption and dAdo, 2,6DCP-dR or 6C2FP-dR formation are calculated. Did the authors control the amount of subproducts not detectable by HPLC?

2. The purity of the products obtained should be stated.

3. The structures of the products were not validated by means other than HPLC retention time. Their NMR validation should be discussed.

4. Authors should explain to the readers, why conversion percentages of thymidine and dAdo remain constant during the process (Figure 3A). It is difficult to understand taking into account that thymidine is being consumed and dAdo formed.

5. Some specialist terms, like “(hydraulic) residence time”, the ??, ?? and ?? tuning parameters of the PID controller (equation 2), and the “environmental or E-factor”, should be also explained to the reader, to avoid that the article is inteligible only by specialists.

6. Potential problems of solubility of substrates and products should be discussed, particularly with the little soluble thymine subproduct and adenine acceptor substrate. For example, they use 10 mM adenine which, according to its molecular mass, is 1.35 g/L, while its solubility in water is around 1 g/L or less (check Merck Index or CRC Press Handbook of Biochemistry and Molecular Biology, or just google for it). Turbidity develops and membrane clogging occurs during the long EMR incubations. The nature of the precipitate was analyzed by SDS-PAGE, which reveals some protein precipitate (Figure S2). However, the authors do not mention whether the insoluble material contained nucleoside and nucleobase material.

7. The two first screenshots of Figure 2B show periods of similar length and similar masses. Are they successive periods, so that the second screenshot corresponds to time 600-1200 s? This should be clarified in the legend or corrected if necessary.

8. The supplementary material contains four figures, but only two of them are mentioned in the main manuscript. In addition, they are wrongly cited as Figure A1 and A2

MINOR ISSUES

9. In the legend to Figure 1, it may be interesting to briefly explain the need/advantage of using two phosphorylases rather than only one.

10. The scheme of Figure 2A shows one EMR setup. In the main text, the use of two parallel EMR systems is reported (lines 116 and 260). This should be also stated in the legend to Figure 2A.

11. In lines 121-123, it is stated that After less than five minutes from the start of the EMR, accurate control of permeate flux was achieved with only minimal overshooting in the beginning (Figure 2B).”. In fact, in the first panel of Figure 2B there is still fluctuation after 300 s, so it seems that accurate control took somewhat longer than the stated 5 min.

12. The heading of the fifth column in Table 1 needs rewriting. The same occurs with the 26DCP-dR abbreviation (missing comma) in the fifth row of the “Product” column.

13. In the head of Table 1, Figure 4 is cited. There is no such figure as, in fact there are two Figures 3. The second one (page 7) must be Figure 4.

14. In line 307, it is stated that “The respective code…”. What is this referring to?

15. Lines 353-354. The use of em dashes is confusing. The intended meaning should be more explicit. In addition, product concentration cannot be expressed as just “mg” (possibly mg/L).

16. An Abbreviations list is missing.

Author Response

Dear reviewer,

we have received your comments and wish to thank you for your time and effort in reviewing our work. Below, we have attached a point by point answer to all your comments. If applicable the suggestions were incorporated into the manuscript and clarification was provided as requested.

  1. In this manuscript the authors describe a continuous procedure to synthesize modified nucleosides using nucleoside thermostable phosphorylases in an enzyme membrane reactor (EMR). In their reactions, thymidine is the sugar-donor substrate and acceptor nucleobases are adenine, and the halogenated-adenine derivatives 2,6-dichloropurine (2,6DCP) and 6-chloro-2-fluoropurine (6C2FP). The synthesis of the corresponding deoxynucleosides (dAdo, 2,6DCP-dR and 6C2FP-dR) was obtained in a one-pot, two-step enzymatic reaction. In the first one, pyrimidine nucleoside phosphorylase catalyzes the phosphorolysis of thymidine yielding thymine and 2’-deoxyribose-1-phosphate. In the second step, purine nucleoside phosphorylase catalyzes the glycosylation of 2’-deoxyribose-1-phosphate by one of the mentioned nucleobases, with liberation of phosphate. Globally, the bi-enzymatic process catalyzes a transglycosylation. So, the deoxynucleoside thymidine is converted to a deoxypurine compound, either dAdo, 2,6DCP-dR or 6C2FP-dR.The authors are based in previous studies of bi-enzymatic batch synthesis of the same or similar compounds with phosphorylases, and in studies of the EMR use for the production of other types of compounds. Both kinds of previous expertise are here conjugated to compare the efficiency of phosphorylase transglycosylations by an EMR procedure versus similar reactions performed in a batch system. The aim of the authors is mainly demonstrative of EMR applicability for the synthesis of nucleoside derivatives, in which they succeed. The manuscript will be relevant for those interested in the production of nucleoside analogs of pharmaceutical interest at industrial scale. However, there are several issues that need revision.

Answer to point 1. Thank you for the positive assessment of our work. Below, we have attached a point by point answer to all your comments and suggestions.

  1. The products formed are analyzed by HPLC monitored at 260 nm. This should detect substrates and (sub)products containing a nitrogen base, but not 2’-deoxyribose-1-phosphate and inorganic phosphate. However, no chromatogram is shown to illustrate the progress and endpoint of substrate consumption and (sub)product formation. This should be presented and used to explain how the percentages of thymidine consumption and dAdo, 2,6DCP-dR or 6C2FP-dR formation are calculated. Did the authors control the amount of subproducts not detectable by HPLC? To calculate conversion percentages first measured peak areas were converted in concentrations based on standard curves obtained from HPLC measurements using defined concentrations of the pure compounds (substrates and products). Afterwards conversion percentages were calculated based on Formula 6 (line 445). We specified that point in Lines 426-429 and Lines 440 to 443: Furthermore, an additional figure (S5) was added to the supplementary material to present exemplary chromatograms (Lines 448-449).

Answer to point 2. Transglycosylation reaction show a balanced interchange of the sugar moiety between the donor nucleoside and the acceptor (in our case a base) catalyzed by a PyNP and PNP. 2’-deoxyribose-1-phosphate released after donor nucleoside cleavage by PyNP is immediately used by the PNP to form the nucleoside of interest. During this reaction inorganic phosphate is released which is directly used for donor nucleoside cleavage. Hence, subproducts are not freely available in the reaction and can therefore not be detected in the reaction. We specified this in Lines 433 to 439.

  1. The purity of the products obtained should be stated.

Answer to point 3. The obtained products were not purified as the focus of the study was on the process design and comparison between process parameters of batch and continuous experiments. Indeed, in a next step it would be an option to connect EMR operation to product purification to validate if product purification is enhanced using EMRs.

Additional text was added to the manuscript to make clear that the products were not purified in the recent study (see lines 142 to 143).

  1. The structures of the products were not validated by means other than HPLC retention time. Their NMR validation should be discussed.

Answer to point 4. For the assignment of the peaks, the retention times and the spectra were compared with those of standards. Further analyses were not performed, since the NMR spectra of the compounds were determined before in the study to determine the biological activity of dihalognated nucleosides (see DOI: 10.3390/molecules25040934).

The text has been slightly adjusted to clarify the identification of the peaks (see lines 428 to 430).

  1. Authors should explain to the readers, why conversion percentages of thymidine and dAdo remain constant during the process (Figure 3A). It is difficult to understand taking into account that thymidine is being consumed and dAdo formed.

Answer to point 5. In the EMR reactions are first run in batch mode. Here, the thymidine concentration decreases, and the product concentration increases until thermodynamic equilibrium is reached (see Figure S1). In the thermodynamic equilibrium the rates of thymidine cleavage and target nucleoside synthesis are equal and no apparent change in concentrations is observed anymore. Only then the continuous mode was started. Here it was important, that the residence time (HRT) in the reactor was long enough so that there was sufficient time for thermodynamic equilibrium to be established. For dAdo formation, the HRT had to be adjusted regularly during the course of the experiment, and for the dihalogenated compounds, a longer residence time in the reactor was already necessary at the beginning.

Additional information was added to better explain this topic (see lines 147 to 150; 167 to 170; 171 to 173; 188 to 190).

  1. Some specialist terms, like “(hydraulic) residence time”, the ??, ?? and ?? tuning parameters of the PID controller (equation 2), and the “environmental or E-factor”, should be also explained to the reader, to avoid that the article is inteligible only by specialists.

Answer to point 6. Thank you very much for this comment. We have added additional explanations to the manuscript.

Lines 319 to 325: “residence time” was changed to “hydraulic residence time” and “τ” was changed to “HRT”. Furthermore, a definition for HRT was given.

Line 85 to 86: We added a short explanation for the E-factor: “[…] E-factor values (ratio of the mass of waste per mass of product) […]”.

Line 339 to 340: The meaning of P, I and D was added.

  1. Potential problems of solubility of substrates and products should be discussed, particularly with the little soluble thymine subproduct and adenine acceptor substrate. For example, they use 10 mM adenine which, according to its molecular mass, is 1.35 g/L, while its solubility in water is around 1 g/L or less (check Merck Index or CRC Press Handbook of Biochemistry and Molecular Biology, or just google for it). Turbidity develops and membrane clogging occurs during the long EMR incubations. The nature of the precipitate was analyzed by SDS-PAGE, which reveals some protein precipitate (Figure S2). However, the authors do not mention whether the insoluble material contained nucleoside and nucleobase material.

The solubility referred to is given for 25°C. However, we applied reaction temperatures of 40°C which allowed to achieve higher solubility.

A sample of the precipitate was also analyzed by HPLC but did not show any peaks corresponding to the nucleoside or nucleobase. The information was added to the text (see line 201 to 204)

  1. The two first screenshots of Figure 2B show periods of similar length and similar masses. Are they successive periods, so that the second screenshot corresponds to time 600-1200 s? This should be clarified in the legend or corrected if necessary.

Answer to point 8. We clarified the description of Figure 2B.

  1. The supplementary material contains four figures, but only two of them are mentioned in the main manuscript. In addition, they are wrongly cited as Figure A1 and A2.

Answer to point 9. Thank you for this comment. We have added the missing references and corrected the names.

  1. In the legend to Figure 1, it may be interesting to briefly explain the need/advantage of using two phosphorylases rather than only one. Transglycosylation is most common one pot reaction and needs both phosphorylases to reach the product from cheap substrate to product ???           

 Answer to point 10. When setting up transglycosylation reactions we pursue 2 goals:

  1. The substrates used should be as cheap as possible.
  2. The product yield should be as high as possible.

For the synthesis of purine nucleoside analogs, both points can be achieved by starting with pyrimidine nucleosides as donors, because:

Referring to 1. Pyrimidine nucleosides are usually much cheaper than purine nucleosides (e.g. adenosine or guanosine derivatives).

Referring to 2. Thermodynamically, pyrimidine nucleosides are preferred donors, whereas for purine nucleosides synthesis is clearly preferred.

We added additional information to the legend of Figure 1.

  1. The scheme of Figure 2A shows one EMR setup. In the main text, the use of two parallel EMR systems is reported (lines 116 and 260). This should be also stated in the legend to Figure 2A.

Answer to point 11. The legend to Figure 2 was re-phrased.

The addition of another reactor in scheme A was not done to keep the scheme clear and simple as the connections do not change.

  1. In lines 121-123, it is stated that “After less than five minutes from the start of the EMR, accurate control of permeate flux was achieved with only minimal overshooting in the beginning (Figure 2B).”. In fact, in the first panel of Figure 2B there is still fluctuation after 300 s, so it seems that accurate control took somewhat longer than the stated 5 min.

Answer to point 12. As the description to Figure 2B was re-phrased (see answer to 7.) it should now be more traceable that in the end of optimization the fluctuations remained in narrow boundaries around the setpoint without the controller overshooting within 5 minutes.

  1. The heading of the fifth column in Table 1 needs rewriting. The same occurs with the 26DCP-dR abbreviation (missing comma) in the fifth row of the “Product” column.

Answer to point 13. The heading was re-written and the missing comma was corrected. Thank you.

  1. In the head of Table 1, Figure 4 is cited. There is no such figure as, in fact there are two Figures 3. The second one (page 7) must be Figure 4.

Answer to point 14. The figures are now correctly labeled. Thank you.

  1. In line 307, it is stated that “The respective code…”. What is this referring to?

Answer to point 15. The paragraph describes the PID-controller which was written as a Python script. In Line 347 we corrected “code” to “Python script” to specify the meaning.

  1. Lines 353-354. The use of em dashes is confusing. The intended meaning should be more explicit. In addition, product concentration cannot be expressed as just “mg” (possibly mg/L).

Answer to point 16. The concentration was indeed missing the volume. “L-1” was added. The em dashes were replaced.

  1. An Abbreviations list is missing.

Answer to point 17. We were not aware that a list of abbreviations was needed. We have now added these to the manuscript.

Reviewer 2 Report

The present manuscript is focused on developing enzyme-mediated production of nucleoside analogs in a continuous enzyme membrane reactor using immobilized nucleoside phosphorylase. The manuscript sounds very interesting, however, some comments need to be addressed before acceptance.

As a major comment, the authors need to explain how EMR is designed and developed, more specifically regarding enzyme immobilization. The reader needs more information about the immobilization process.  In this regard, I recommend the authors include an additional chapter in the results and discussion subsection (and in materials and methods too) including detailed information about the enzyme immobilization process.

Additionally, the manuscript displays a lot of interesting results but needs a more detailed discussion about the differences obtained by using the different systems (EMRs and batch reactions)

Minor comments:

Lines 64-65 Please, rewrite the phrase: “In recent years, thermostable NPs have become increasingly interesting (¿catalysts?)………

Line 109: The constants (Kp, KI, KD) must be written in italics 

Author Response

Dear reviewer,

we have received your comments and wish to thank you for your time and effort in reviewing our work. Below, we have attached a point by point answer to all your comments. If applicable the suggestions were incorporated into the manuscript and clarification was provided as requested.

  1. The present manuscript is focused on developing enzyme-mediated production of nucleoside analogs in a continuous enzyme membrane reactor using immobilized nucleoside phosphorylase. The manuscript sounds very interesting, however, some comments need to be addressed before acceptance.

Answer to point 1. Thank you for the positive assessment of our work.

  1. As a major comment, the authors need to explain how EMR is designed and developed, more specifically regarding enzyme immobilization. The reader needs more information about the immobilization process. In this regard, I recommend the authors include an additional chapter in the results and discussion subsection (and in materials and methods too) including detailed information about the enzyme immobilization process.            

Answer to point 2. In this study we decided to us an enzyme membrane reactor to avoid enzyme immobilization The enzymes are retained in the reactor by choosing a membrane with a molecular weight cut-off smaller than the enzymes, so that the freely dissolved enzymes are just held back in the reactor.

To make this even clearer, we have added an additional sentence to the abstract (see Lines 25 to 26).

  1. Additionally, the manuscript displays a lot of interesting results but needs a more detailed discussion about the differences obtained by using the different systems (EMRs and batch reactions).

Regarding the differences of batch and continuous reactions we compared the product yields per enzymatic unit that increased by a factor of 9 and 40 for the modified nucleosides and even 220 for the natural substrate when the continuous reactor was used (Line 263 and Line 194). In addition, we could show the longevity of the NPs under reaction conditions which was comparable to other studies in flow reactors (Lines 183 – 185).

  1. Lines 64-65 Please, rewrite the phrase: “In recent years, thermostable NPs have become increasingly interesting (¿catalysts?)………

Answer to point 4. The sentence was re-phrased.

  1. Line 109: The constants (Kp, KI, KD) must be written in italics

Answer to point 5. Thank you. We corrected the constants to be written in italics.

Round 2

Reviewer 1 Report

All the queries posed by this reviewer have been attended satisfactorily, except for a few minor mistakes that remain uncorrected or appear in the new material added

New Figure S5. Numerical data for conversions in part A show less significant figures than in parts B and C. They should be all the same. In the legend “dAdo-“ should be “dAdo”.

In line 89, “advantages” should be “advantageous”

In line 202, “did to lead” shoud be “did not lead”

In line 250, “Figure 3” should be “Figure 4”

In line 42, I think the Abbreviation list is out of place according to the uses of the journal. Usually it goes at the end, together with Author contributions, etc

Author Response

Dear reviewer,

we wish to thank you again for your time and effort in having a second look into our work. Below, we have attached a point by point answer to your comments.

  1. All the queries posed by this reviewer have been attended satisfactorily, except for a few minor mistakes that remain uncorrected or appear in the new material added.

Answer to point 1. We are pleased to hear that all comments were attended satisfactorily. The remaining mistakes were now corrected (see below).

  1. New Figure S5. Numerical data for conversions in part A show less significant figures than in parts B and C. They should be all the same. In the legend “dAdo-“ should be “dAdo”.

Answer to point 2. In Figure S5, all numbers now only show one decimal digit. In the legend “dAdo-“ was changed to “dAdo”.

  1. In line 89, “advantages” should be “advantageous”

Thank you. “advantages” was changed to “advantageous” (see line 73)

  1. In line 202, “did to lead” shoud be “did not lead”

Answer to point 4. Thank you. “did to lead” was changed to “did not lead” (see line 186)

  1. In line 250, “Figure 3” should be “Figure 4”

Answer to point 5. We apologize that Figure 4 continued to be mislabeled in the pdf version of the manuscript. We have now corrected this.

  1. In line 42, I think the Abbreviation list is out of place according to the uses of the journal. Usually it goes at the end, together with Author contributions, etc

Answer to point 6. The abbreviations are now at the end of the manuscript.

Kind regards,

Anke Kurreck, on behalf of all authors